# Infrared Spectroscopy of Be Stars: Influence of the Envelope Parameters on Brackett-Series Behaviour

Yanina Roxana Cochetti [1,2,*], Anahi Granada [3], María Laura Arias [1,2], Andrea Fabiana Torres [1,2] and Catalina Arcos [4]

1 Instituto de Astrofísica de La Plata (CCT La Plata—CONICET, UNLP), Paseo del Bosque S/N, La Plata B1900FWA, Argentina
2 Departamento de Espectroscopía, Facultad de Ciencias Astronómicas y Geofísicas, Universidad Nacional de La Plata, La Plata B1900FWA, Argentina
3 Centro Interdisciplinario de Telecomunicaciones, Electrónica, Computación y Ciencia Aplicada (CITECCA), Sede Andina, Universidad Nacional de Río Negro, San Carlos de Bariloche R8400AHN, Argentina
4 Instituto de Física y Astronomía, Facultad de Ciencias, Universidad de Valparaíso, Valparaíso 2360102, Chile
* Correspondence: cochetti@fcaglp.unlp.edu.ar

**Abstract:** The IR spectra of Be stars display numerous hydrogen recombination lines, constituting a great resource for obtaining information on the physical and dynamic structures of different regions within the circumstellar envelope. Nevertheless, this spectral region has not been analysed in depth, and there is a lack of synthetic spectra with which to compare observations. Therefore, we computed synthetic spectra with the HDUST code for different disc parameters. Here, we present our results on the spectral region that includes lines of the Brackett series. We discuss the dependence of the line series strengths on several parameters that describe the structure of the disc. We also compared model line profiles, fluxes, and EWs with observational data for two Be stars (MX Pup and $\pi$ Aqr). Even though the synthetic spectra adequately fit our observations of both stars and allow us to constrain the parameters of the disc, there is a discrepancy with the observed data in the EW and flux measurements, especially in the case of MX Pup. It is possible that by including Brackett lines of higher terms or adding the analysis of other series, we may be able to better constrain the parameters of the observed disc.

**Keywords:** stars: early-type; stars: emission-line, Be; stars: individual: MX Pup, $\pi$ Aqr; circumstellar matter

## 1. Introduction

Classical Be stars are non-supergiant B-type stars whose spectra show or have shown the H$\alpha$ line in emission [1,2]. These stars are rapidly rotating, and it is accepted that this emission originates in a circumstellar envelope, mostly compatible with a disc geometry in Keplerian rotation [3]. The model that best explains the observations of Be stars is the viscous decretion disc [4], which allows us not only to explain static discs but also to study their dynamical evolution [5].

In the infrared (IR) spectral region, Be spectra present a moderate flux excess and numerous hydrogen emission lines of the Paschen, Brackett, Pfund, and Humphreys series. Since these IR lines are formed in a region closer to the star than those observed in the optical spectral range, and the photospheric absorption is almost negligible for most of them, the study of these IR emission lines is a great tool to obtain information about the physical structure and dynamics within the innermost part of the disc [6–10].

This spectral region has gained importance in recent decades. An extensive atlas of the K- and L-bands of early-type stars, including Be stars, was published by Lenorzer et al. [11]. With those data, Lenorzer et al. [7] made a diagram with the flux ratios Hu14/Br$\alpha$ versus Hu14/Pf$\gamma$. In this diagram, the location of the objects depends on the density of the

emitting gas. Mennickent et al. [8] proposed a classification criterion based on the emission lines observed in the L-band: those stars on which the Humphreys lines present intensities similar to those of Br$\alpha$ and Pf$\delta$ lines constitute Group I; stars with Br$\alpha$ and Pf$\delta$ lines more intense than Humphreys lines are part of Group II; and stars without emission lines are part of Group III. Groups I and II fall in different regions of Lenorzer's diagram; thus, the group membership is probably connected to the density of the disc. Classifying a sample of Be stars using Mennickent's criterion, Granada et al. [9] found that the equivalent widths (EWs) of Br$\alpha$ and Br$\gamma$ lines are similar for Group I objects, while for stars in Group II, EW(Br$\gamma$) is much larger (more than five times) than EW(Br$\alpha$).

Up to the recent publication of Cochetti et al. [12]'s atlas, most of the works that focused on the IR data of Be stars showed low-resolution data, analysed a small sample, or covered a small spectral range. The already-mentioned atlas comprises medium-resolution spectra in the near-IR region of a sample of Galactic Be stars. By measuring different parameters of the observed hydrogen recombination lines, these authors diagnosed the physical conditions in the circumstellar environment. They defined new complementary criteria to classify Be stars according to their disc opacity. This atlas provides valuable observational material available to be analysed and modelled. In particular, the behaviour of the fluxes of the hydrogen lines for different stars may give us clues about the properties of the discs. To perform better analysis, modelling synthetic spectra and comparing them with observations is essential.

In the last years, different observables from Be star discs have been successfully modelled in the framework of the viscous decretion disc model (VDD; [4,13]). In this model, the rapidly rotating central star continuously loses mass and angular momentum and puts material in a Keplerian orbit at the base of the disc. Beyond this inner edge, the material is expected to be driven outwards through some turbulent viscous process. The rotationally distorted star irradiates the disc, which is assumed to extend out to a radius of $R_d$ equatorial radii, and the star-plus-disc system is viewed at an inclination angle (angle between the star's rotation axis and the line of sight) of $i$. Then, in state-of-the-art disc models such as HDUST [14] or Bedisc/BeRay [15,16], each computed observable is a function of four model parameters: $\rho_0$, $n$, $R_d$, and $i$, where $\rho_0$ is the axisymmetric volume density at the innermost part of the disc, and $n$ is the exponent of the radial power law usually used to describe the Be star disc's density.

The modelling of different observables has allowed constraints on Be disc parameters to be derived. In particular, modelling of the iconic H$\alpha$-line emission has shown that this line typically forms within the innermost tens of stellar radii [3,17,18]. The modelling of the Br$\gamma$ line, in particular, from spectro-interferometric data [19] has allowed for constraining stellar parameters and also shown that the formation region of this line is, at most, 15 or 20 stellar radii. However, a smaller Brackett-series-forming region of a few stellar radii has been derived from near-IR spectra [20]. Unfortunately, the lack of simultaneous observations prevents a direct comparison between H$\alpha$- and Br$\gamma$-line-forming regions. The modelling of the near-IR hydrogen recombination lines allows us to explore the properties of their line-forming region, which is expected to be closer to the star than that of H$\alpha$ and, thus, contributes to better constraints on the parameters of the disc.

With this aim, we started to compute a grid of synthetic spectra covering the near-IR spectral range with the HDUST code [14,21]. As a first step, we computed the first lines of the Brackett series for a set of disc parameters and one central B-type star. Even though we do not expect to be able to describe the whole complexity of the star–disc interphase, we seek to understand the impact on the emerging spectra of the different parameters involved in the modelling of Be stars in the viscous decretion disc framework, with the final goal of deducing disc properties from the IR spectra. From the synthetic spectra, we measured the equivalent widths and fluxes of the hydrogen lines and analysed the behaviour of the line series when varying the parameters that describe the disc density law. Finally, we compared the spectra and the line flux behaviour from the models with the observational data of two known Be stars: MX Pup and $\pi$ Aqr.

## 2. Methods

We used the HDUST code, developed by Carciofi and Bjorkman [14,21]. This three-dimensional non-LTE Monte Carlo code solves the radiative transfer, radiative equilibrium, and statistical equilibrium equations in a 3D geometry for different gas density and velocity distributions. This code has been used to discuss the properties of Be stars via the analysis of the spectral energy distribution (SED) based on different wavelengths, as well as some optical hydrogen lines [22–25].

The code includes a 25-level model for the hydrogen atom, of which the first 12 are explicit non-LTE levels, and the upper 13 are implicit levels with the LTE population. This default configuration of the code allowed us to compute the Brackett series from the Br$\alpha$ line to Br12. We also modelled the continuum in the range 1.4–4.3 $\mu$m.

The disc is considered to start at the stellar radius $R_\star$, and the density throughout it is modelled with the expression

$$\rho(r,z) = \rho_0 \left( \frac{r}{R_\star} \right)^{-n} exp\left( -\frac{z^2}{2H^2} \right) \tag{1}$$

which includes a power law and a Gaussian distribution in the radial and vertical directions, respectively [14]. In this expression, $\rho$ is the density at each pair of $(r,z)$ coordinates, $\rho_0$ is the density at the base of the disc, and $H$ is the scale height of the disc. For a vertically isothermal disc, $H$ is obtained with the expression

$$H = H0 \left( \frac{r}{R_\star} \right)^{1.5}, H0 = \sqrt{\frac{kT}{\mu \cdot m_H}} \frac{1}{V_{orb}} R_\star \tag{2}$$

where $k$ is the Boltzmann constant, $\mu$ is the mean molecular weight, $m_H$ is the mass of the hydrogen atom, $V_{orb} = \sqrt{G\,M/R_{eq}}$ is the disc orbital speed at the stellar equator $R_{eq}$, and the temperature $T$ has been set as 72% of the effective temperature $T_{\text{eff}}$ of the star.

Despite the evidence of differential rotation reported in the literature (i.e., [26]), to date, there is not a comprehensive theory that covers all the observed physical phenomena occurring in the interior and photosphere of rapidly rotating stars. Then, the adoption of a solid and rigid approach is prevalent in their modelling, and particularly in this code.

We calculated models with $\rho_0$ values between $5 \times 10^{-12}$ and $1 \times 10^{-10}$ g cm$^{-3}$[1] and values of the exponent $n$ in the range 2.5–4 [23]. These $\rho_0$ values correspond to a steady-state decretion rate $\dot{M}$ over the viscosity parameter $\alpha$ [27] in the range $2.13 \times 10^{-13} M_\odot$ yr$^{-1} \lesssim \dot{M}/\alpha \lesssim 2.13 \times 10^{-11} M_\odot$ yr$^{-1}$[23]. Following a procedure similar to the one described by Silaj et al. [28] for determining the convenient disc size for each $n$–$\rho_0$ pair, we analysed the change in the equivalent width of the modelled lines as a function of the disc size by increasing it by 10 $R_\star$ each time. Then, we determined the minimum disc size at which the EW reached a stable value for each model. The required radius is larger for lower $n$ values or higher central densities. By using this radius as the disc's outer radius $R_d$, we ensure that we include the entire emitting region. To check the convergence of the simulation, we plotted the disc temperature for the last iterations, ensuring a decreasing profile in the inner part of the disc and an increasing temperature in the outer disc (forming a typical U shape, [14]).

In Table 1, we show the envelope's outer radius $R_d$ chosen for each $n$–$\rho_0$ combination employed in the modelling[2]. We computed models that have been found adequate to fit Be-star H$\alpha$ profiles and SEDs and are related to different dynamical stages of the disc: dissipating discs present $n \lesssim 3$, while the range $3 \lesssim n \lesssim 3.5$ is for stable discs, and $3.5 \lesssim n$ is related to discs in formation [23]. Models with a small exponent and high density $\rho_0$ trace the "forbidden zone", defined by Vieira et al. [23] as a region in the $n$–$\rho_0$ diagram with unobserved combinations of these parameters. Reaching this region would require very massive circumstellar discs.

As the first step for this work, we ran simulations for only one set of fundamental parameters for the central stellar model. We used a self-consistent rigid rotator with the following parameters: a mass $M = 10\,M_\odot$, a polar radius $R_{pole} = 5.5\,R_\odot$, a luminosity $L = 7500\,L_\odot$, and a rotational rate $W = 0.7$. We recall that $W = V_{rot}/V_{orb}$, where $V_{rot}$ is the rotational velocity at the equator of the star, and $V_{orb}$ is the equatorial circular orbital (Keplerian) velocity, as defined by Rivinius et al. [3]. This set of parameters corresponds to an early B star with $T_{eff} \simeq 24{,}250\,\mathrm{K}$ and $V_{rot} \simeq 370\,\mathrm{km\,s^{-1}}$. For producing synthetic profiles with HDUST, the gravity-darkening effect is accounted for, considering a parameter $\beta = 0.188$ suitable for rapidly rotating stars [29]. As it has been proposed that late-type Be stars could have more tenuous discs than early-type Be stars due to the smaller variability observed [3,10], in a forthcoming article, we plan to include central stars with different spectral types to analyse the dependence on the disc parameters.

To obtain the synthetic spectra and measure the line parameters, the results were processed with PyHdust[3] and specutils[4] Python packages.

**Table 1.** Envelope's outer radius $R_d$ for each $n$ and $\rho_0$ combination. According to Vieira et al. [23], the $n$ value is related to different disc stages: dissipating discs present $n \lesssim 3$, while the range $3 \lesssim n \lesssim 3.5$ is for stable discs, and $3.5 \lesssim n$ is related to discs in formation. For each model, we simulated the spectra for inclination angles in the range 0–60° with a 15° step.

| $n$ | $\rho_0 = 1 \times 10^{-12}$ $(\mathrm{g\,cm^{-3}})$ | $\rho_0 = 5 \times 10^{-12}$ $(\mathrm{g\,cm^{-3}})$ | $\rho_0 = 1 \times 10^{-11}$ $(\mathrm{g\,cm^{-3}})$ | $\rho_0 = 5 \times 10^{-11}$ $(\mathrm{g\,cm^{-3}})$ | $\rho_0 = 1 \times 10^{-10}$ $(\mathrm{g\,cm^{-3}})$ |
|---|---|---|---|---|---|
| 2.5 | $30\,R_\star$ | × | × | × | × |
| 3.0 | $20\,R_\star$ | $20\,R_\star$ | $30\,R_\star$ | $50\,R_\star$ | × |
| 3.5 | $20\,R_\star$ | $20\,R_\star$ | $20\,R_\star$ | $30\,R_\star$ | $30\,R_\star$ |
| 4.0 | $20\,R_\star$ | $20\,R_\star$ | $20\,R_\star$ | $20\,R_\star$ | $20\,R_\star$ |

## 3. Results

For each simulation, we generated spectra for inclination angles in the range of 0–60°, with a 15° step. In Figure 1, we show as an example the normalised spectrum obtained for the model with $n = 3.5$ and $\rho_0 = 10^{-11}\,\mathrm{g\,cm^{-3}}$, viewed from a pole-on ($i = 0°$) orientation. In the following subsections, we will show the analysis performed for the 0°, 30°, and 60° orientations.

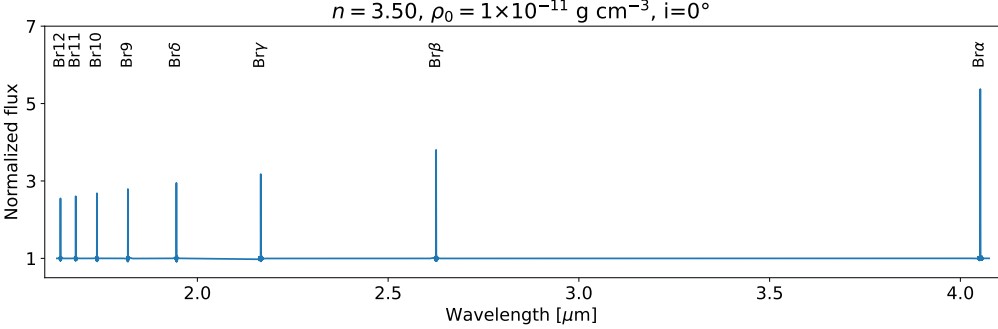

**Figure 1.** Spectrum from the simulation with $n = 3.5$ and $\rho_0 = 10^{-11}\,\mathrm{g\,cm^{-3}}$, with a pole-on ($i = 0°$) orientation. Brackett lines are labelled.

### 3.1. Brackett-Series Behaviour According to n and $\rho_0$

Figures 2–5 show the synthetic line profiles for the Br$\alpha$ to Br12 lines for the different simulations for $i = 0°$. Each figure corresponds to a different value of $n$, and in each plot, the different colours are for different $\rho_0$ values. Synthetic line profiles for $i = 30°$ and $i = 60°$ are in Appendix A.1.

For the behaviour across the series, we observe the following:

- For $n = 2.5$ (Figures 2 and A1), which is the lowest $n$ value and can only be combined with the lowest density ($\rho_0 = 10^{-12}\,\mathrm{g\,cm^{-3}}$), there is a strong intensification for the lines from Br12 to Br$\alpha$ for all the inclinations. The lines are more intense for lower inclinations.
- For $n = 3.0$ (Figures 3 and A2), the height of each line relative to its adjacent continuum also increases from Br12 to Br$\alpha$. The slope of the increase is different for each density: the lower the density, the stronger the increase. For $i = 0°$ and $i = 30°$, the increase is higher than for $i = 60°$.
- For $n = 3.5$ (Figures 4 and A3), the slope of the increase is also steeper for the lowest densities, but not as remarkable as for $n = 3.0$. The higher increase for lower densities means that, even though the higher-order lines are more intense for the highest densities, the first members of the series present similar intensities for intermediate densities.
- For $n = 4.0$ (Figures 5 and A4), the intensities are the smallest, with the increase not so different for each density from Br12 to Br$\alpha$.

Since the $n$ value is related to the dynamical state of the disc (forming, steady, or dissipating [23]), it was expected that different behaviour would be found for the line intensity across the series.

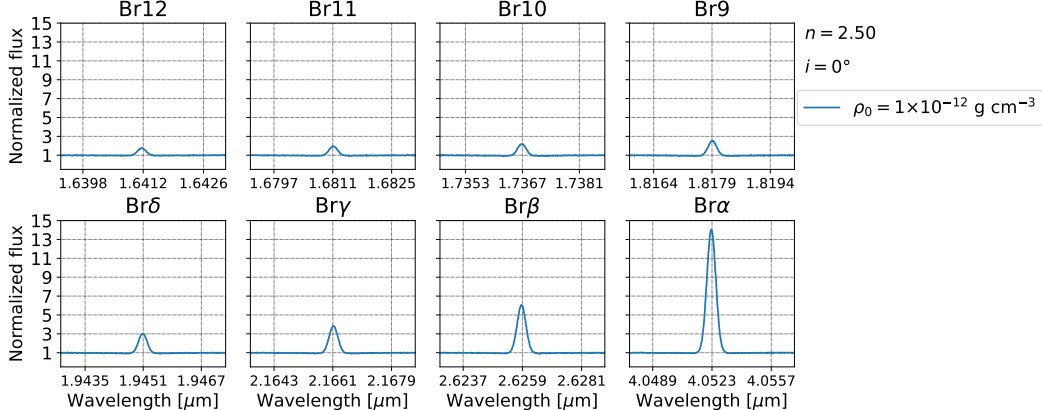

**Figure 2.** Synthetic Brackett line profiles obtained for $i = 0°$, an exponent of the density law $n = 2.5$, and $\rho_0 = 1 \times 10^{-12}\,\mathrm{g\,cm^{-3}}$.

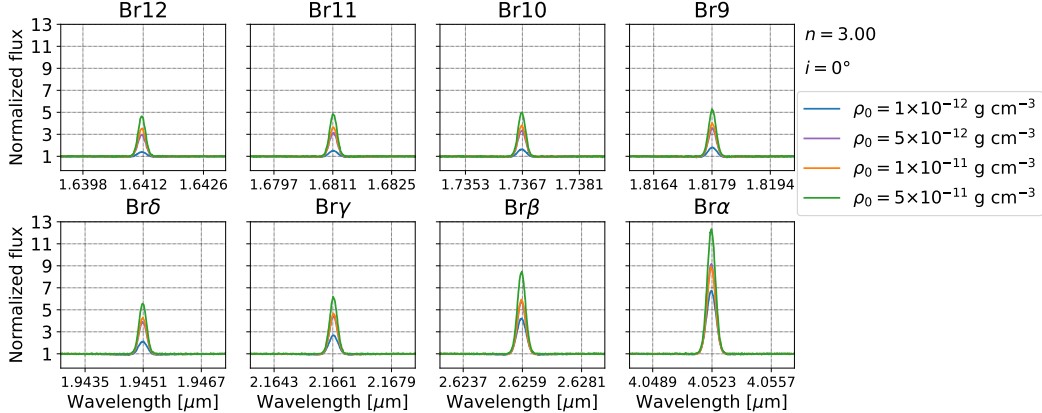

**Figure 3.** Synthetic Brackett line profiles obtained for $i = 0°$, an exponent of the density law $n = 3.0$, and different values of the central density.

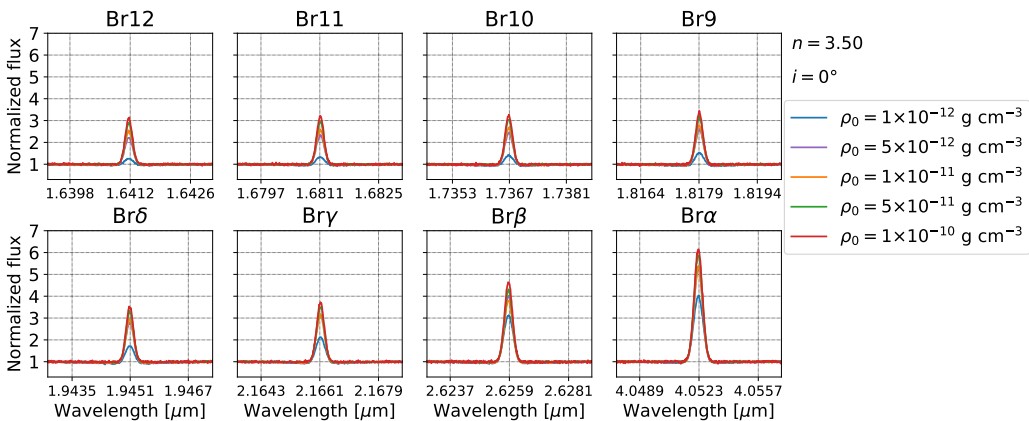

**Figure 4.** Synthetic Brackett line profiles obtained for $i = 0°$, an exponent of the density law $n = 3.5$, and different values of the central density.

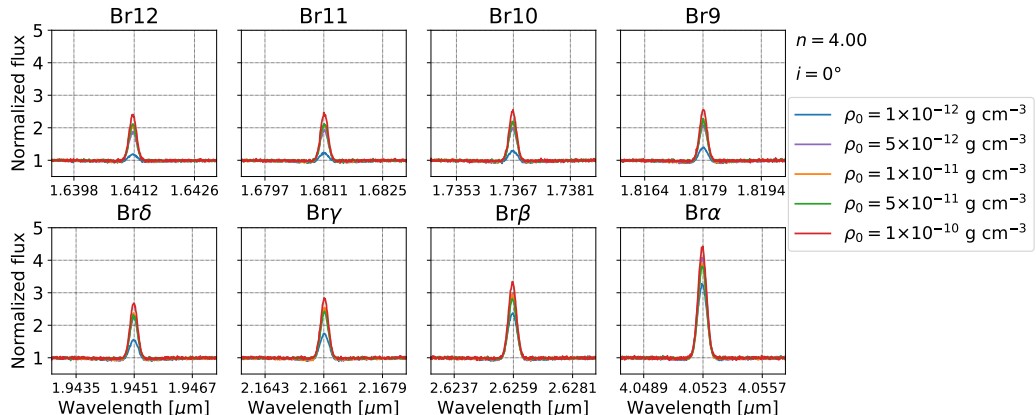

**Figure 5.** Synthetic Brackett line profiles obtained for $i = 0°$, an exponent of the density law $n = 4.0$, and different values of the central density.

### 3.2. EW and Flux Ratios Relative to Br12

To deduce the contribution of the disc, we corrected the measured emission line EWs for photospheric absorption [12]. The corrected EW values for $i = 0°$ for each $n$ and $\rho_0$ normalised to EW(Br12) are plotted in Figure 6. Plots for $i = 30°$ and $i = 60°$ are in Appendix A.2. As we described before for the line profiles, for all the curves, there is an increase in the EW from Br12 to Br$\alpha$, and in each panel (i.e., each density and inclination), the higher the $n$ value, the smaller the slope of the curve.

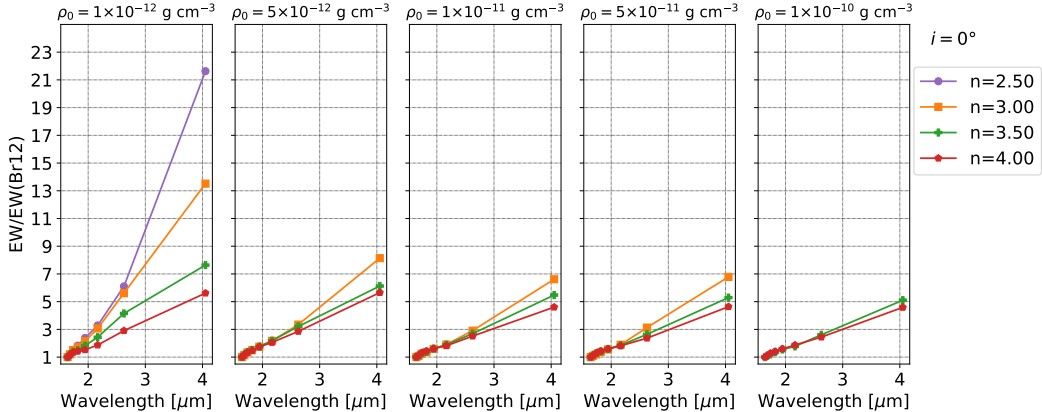

**Figure 6.** EW ratio for the Brackett lines relative to EW(Br12) for $i = 0°$ (for other inclinations see Figure A5). Each panel corresponds to a central density $\rho_0$, and each curve represents a given $n$ value.

We obtained the flux for each line using the corrected line EW and the flux of the continuum at each wavelength. The fluxes normalised to flux(Br12) are plotted in Figure 7 for $i = 0°$ and in Appendix A.2 for $i = 30°$ and $i = 60°$. All the curves present non-monotonic behaviour, with fluxes increasing from Br12 to Br10 and then decreasing up to Br$\alpha$. The lowest curve corresponds to the highest $n$ value in each panel. For the lowest density, the maximum of the curve of line flux ratios is higher for lower $n$, but for the other densities, the maximum value is very similar for all the exponents.

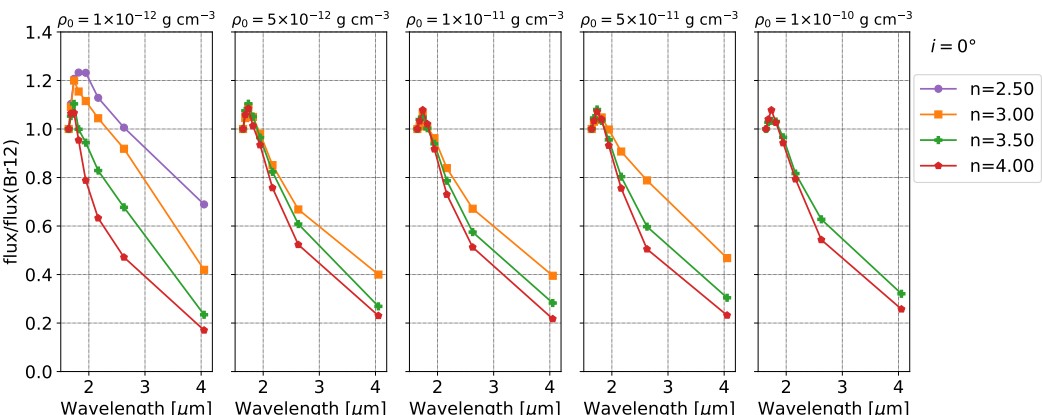

**Figure 7.** Flux ratio for the Brackett lines relative to flux(Br12) for $i = 0°$ (for other inclinations see Figure A6). Each panel corresponds to a central density $\rho_0$, and each curve represents a given $n$ value.

In Figure 8, we show the plot of EW(Br$\alpha$) versus EW(Br$\gamma$). We can see that the models with a higher exponent value present a smaller EW(Br$\alpha$)/EW(Br$\gamma$) ratio and are close to the relation expected for Group I objects [9,12]. The Group II locus seems to only contain objects with envelopes with the smallest density. For each $n$, the higher the density, the higher the EWs.

Finally, Figure 9 shows EW(Br12) versus EW(Br11). This plot shows that the different models we calculated follow a linear relation close to the expected ratio for case B recombination, similarly to that obtained by Steele and Clark [31] (see their Figure 8 , top). As for our Figure 8, larger values of EWs are obtained for smaller $n$ values, and for a fixed $n$ value, the higher the density, the higher the EWs. The behaviour is similar for the different inclination angles computed.

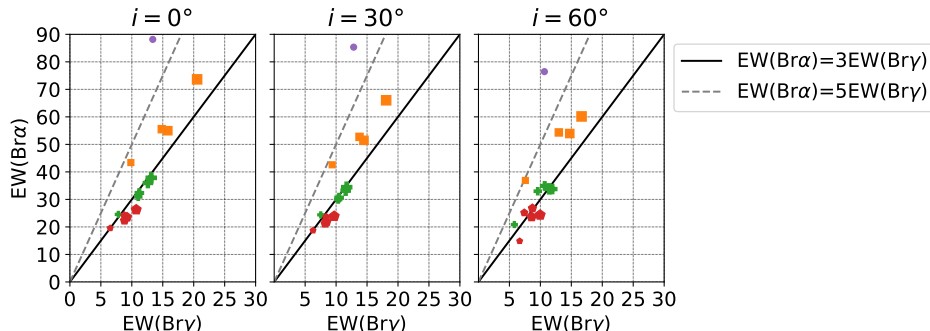

**Figure 8.** EW(Br$\alpha$) versus EW(Br$\gamma$). Colours and symbols are the same as in Figure 7. The symbol size is proportional to the central density of the model.

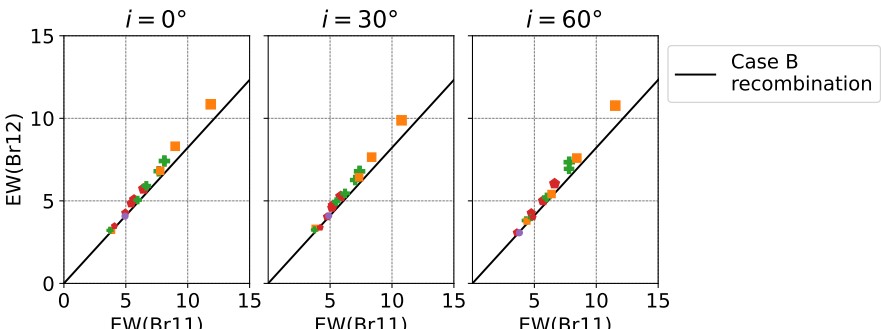

**Figure 9.** EW(Br12) versus EW(Br11). Colours and symbols are the same as in Figure 7. The symbol size is proportional to the central density of the model.

## 4. Discussion

We seek to explore whether the computed models successfully describe the observed data published by Cochetti et al. [12]. With this aim, we first perform Brackett line fitting and then compare the synthetic line ratios with those of the observed spectra. This way, we can analyse the possibility of determining the disc parameters $\rho_0$ and $n$. Using the published sample in Cochetti et al. [12], we have chosen two stars with stellar parameters close to those used in our modelling: MX Pup and $\pi$ Aqr. For these stars, spectra were obtained in June 2017 with the FIRE spectrograph installed at the Magellan Baade Telescope at Las Campanas Observatory.

For the comparison with observational data, we allow the inclinations to span the whole range of our computations, 0°, 15°, 30°, 45°, and 60°. Theoretical spectra have been convolved with a Gaussian profile with FWHM = 60 km s$^{-1}$ to match the observed data's spectral resolution. The uncertainties for the measurements are around 10% for the EW and flux due to possible errors in the continuum determination.

### 4.1. MX Pup (HD 68980)

For MX Pup, Frémat et al. [32] determined an effective temperature $T_{\text{eff}} = 25{,}125 \pm 642$ K and a projected rotational velocity $V \sin i = 152 \pm 8$ km s$^{-1}$. From fitting on the H$\alpha$ profile, Silaj et al. [33] found an inclination angle $i = 20°$, while Arcos et al. [18] determined $i = 50°$ from data taken in 2013 and 2015.

We made a visual comparison between each model and the spectrum of MX Pup, and Figure 10 shows the best-fitting model. On the x-axis, in velocity units, we can see that the peak separation for higher-order members is larger than in Br$\gamma$, indicating that their forming region is closer to the central star. Since HDUST does not include non-coherent electron scattering, which can affect the line profiles with extended wings [34], a difference in the wings still remains, meaning that the non-coherent electron scattering is significant. This could be improved in the future by convolving the model with a Gaussian to simulate the electron velocity [35]. It is also possible that a single power law is not enough to fit the complexity of the disc if the wings and the core are formed in very different regions of the disc [36]. Because of this, as the best-fitting model, we chose a compromise solution for both the central part of the line and the wings. The parameters of the model are $n = 3.0$, $\rho_0 = 5 \times 10^{-11}$ g cm$^{-3}$, and $i = 60°$. Such parameters put this object in the stable/dissipating limit according to Vieira et al. [23]. With this inclination, the rotational velocity would be smaller than the one used for our modelling but inside the range of the true rotational velocity ratios expected for Be stars [37]. In the EW/EW(Br12) and flux/flux(Br12) plots (Figure 11), the observed data agree with the curve for the values determined from the spectrum. Nevertheless, in the EW(Br11) vs. EW(Br12) plot in Figure 12, the MX Pup position lies over the diagonal but far from the models. This difference may be caused by the poor fitting of the line wings.

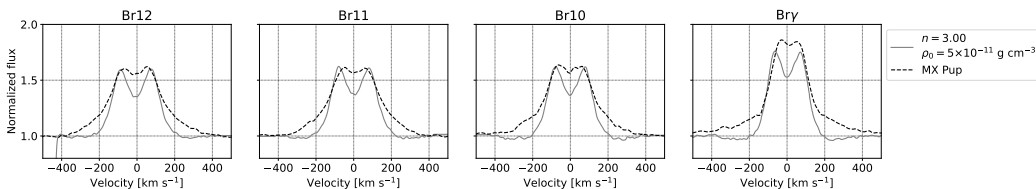

**Figure 10.** Best-fitting model for MX Pup.

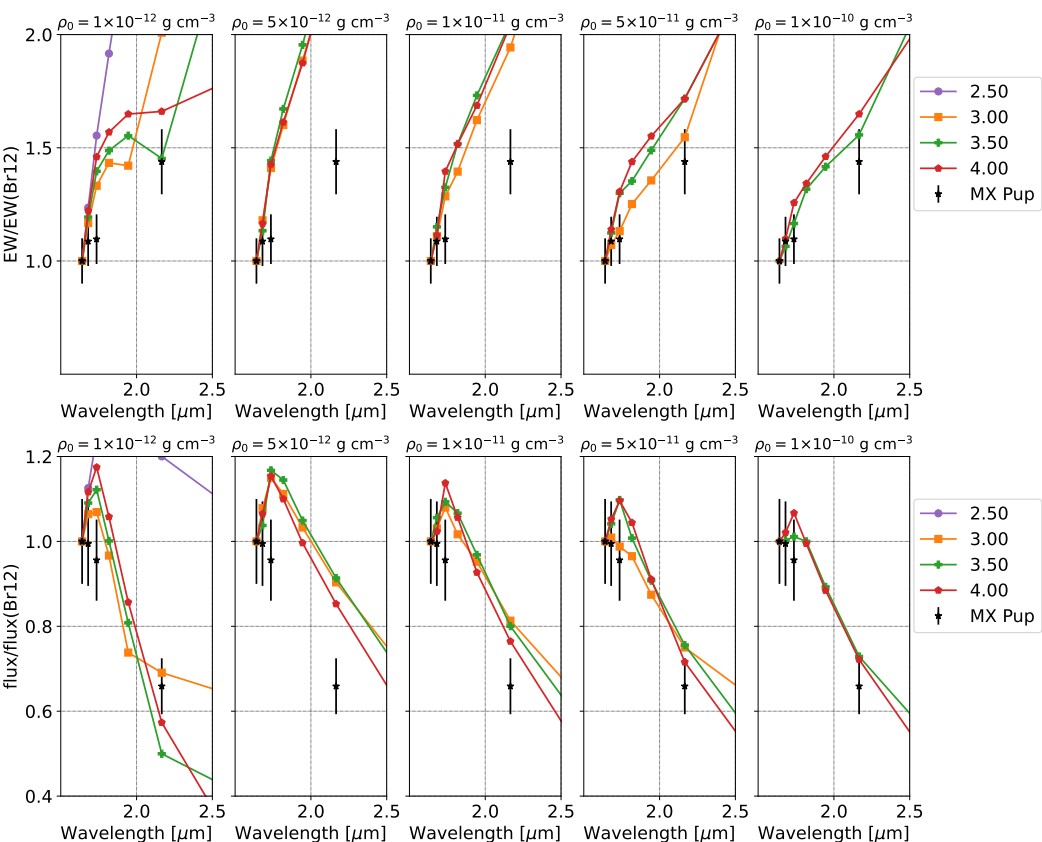

**Figure 11.** Comparison of the curves of EW and flux relative to those of Br12 for $i = 60°$ and different central densities $\rho_0$ and $n$ values with the data obtained for the Be star MX Pup.

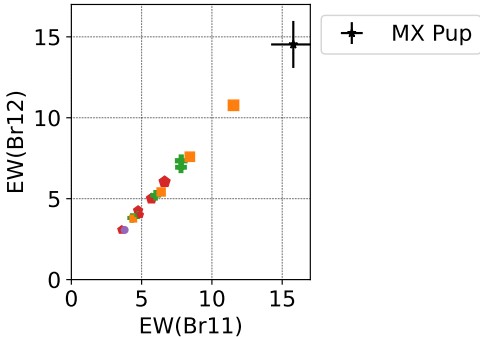

**Figure 12.** Location of the measurements for the star MX Pup in the EW(Br12) versus EW(Br11) plot for $i = 60°$. Colours and symbols are the same as in Figure 7. The symbol size is proportional to the central density of the model.

### 4.2. π Aqr (HD 212571)

For this object, Frémat et al. [32] determined an effective temperature $T_{\text{eff}} = 26{,}061 \pm 736$ K and a projected rotational velocity $V \sin i = 233 \pm 15$ km s$^{-1}$. Both Silaj et al. [33]

and Arcos et al. [18] reported inclination angles for this object, with $i = 45°$ and $i = 60°$, respectively.

The best-fitting model found through a visual comparison is shown in Figure 13. The peak separation for the higher-order lines is not as different from the one for Br$\gamma$ as in MX Pup. In this case, it does not seem to be necessary to add non-coherent scattering to fit the wings. The parameters are the following: $n = 3.5$, $\rho_0 = 10^{-11}$ g cm$^{-3}$, and $i = 45°$. This corresponds to a forming/stable disc [23]. Although in the EW/EW(Br12) plot, $\pi$ Aqr data follow the tendency of the model from the spectral fitting, in the flux/flux(Br12) plot, the behaviour of the observed lines does not follow the model's tendency (Figure 14). Figure 15 shows the EW(Br12) vs. EW(Br11) plot, where $\pi$ Aqr data agree with the parameters found previously.

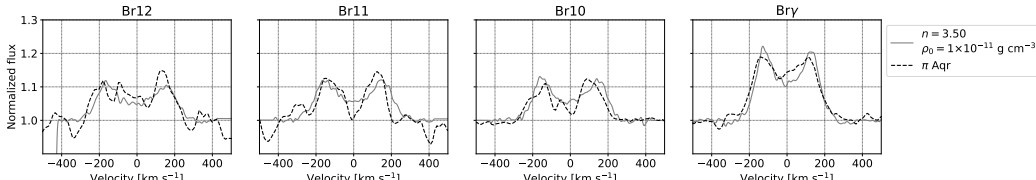

**Figure 13.** Best-fitting model for $\pi$ Aqr.

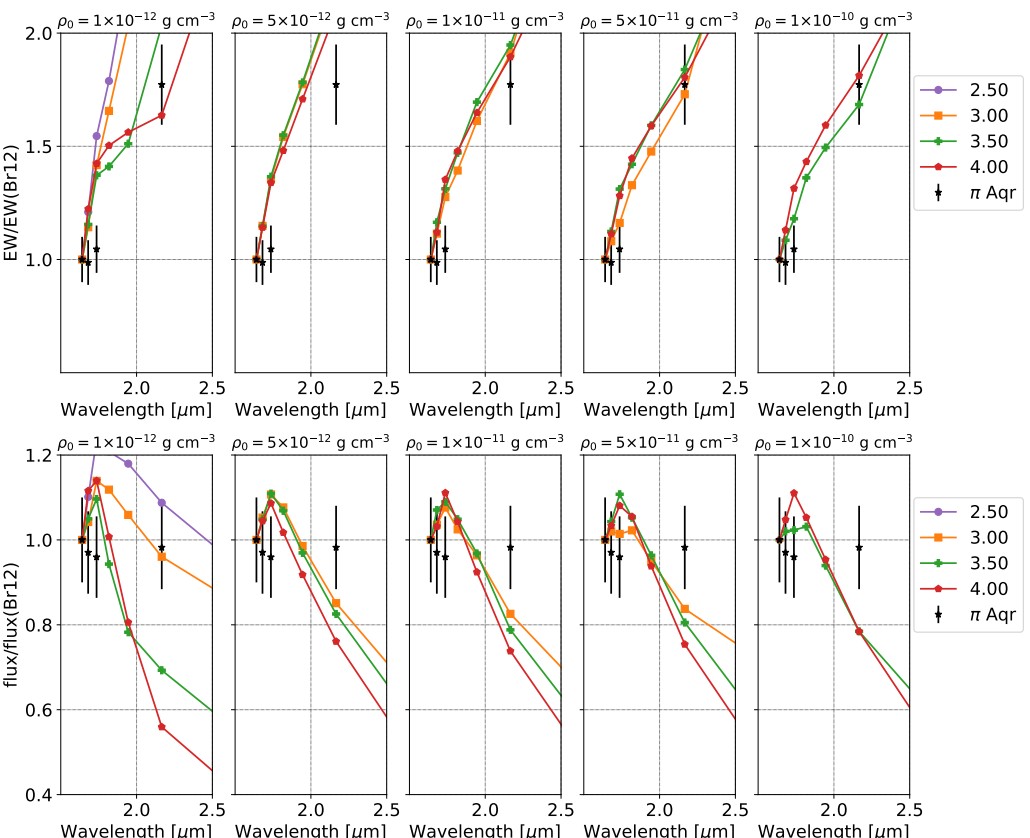

**Figure 14.** Comparison of the curves of EW and flux relative to those of Br12 for $i = 45°$ and different central densities $\rho_0$ and $n$ values with the data obtained for the Be star $\pi$ Aqr.

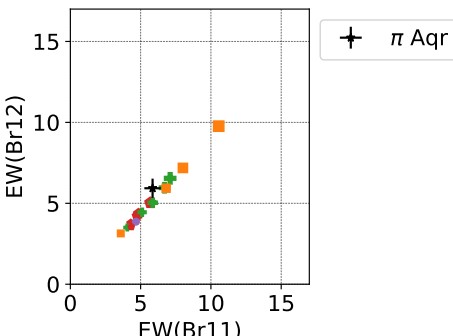

**Figure 15.** Location of the measurements for the star $\pi$ Aqr in the EW(Br12) versus EW(Br11) plot for $i = 45°$. Colours and symbols are the same as in Figure 7. The symbol size is proportional to the central density of the model.

## 5. Conclusions

We used the HDUST code to perform simulations that allowed us to obtain synthetic Brackett-series line profiles for one set of parameters for the central star corresponding to an early-type object, with different parameters for the density law throughout the disc. The calculations of the synthetic models were carried out for different inclination angles of the system (star-plus-disc). The parameter space of the models that we used has been found suitable to fit Be-star H$\alpha$ profiles (e.g., [23]).

After measuring the EW and flux of the Brackett-series line profiles and applying a correction for photospheric absorption, we analysed the behaviour of the series according to the disc parameters. We found that the line intensity increment through the series depends on those parameters. Regarding the EW(Br$\alpha$) versus EW(Br$\gamma$) behaviour, the calculated models lie in different loci according to the central density. In the EW(Br12) versus EW(Br11) plot, the models follow a linear relation close to the expected ratio for case B recombination [31].

By comparing our simulations with the data obtained for MX Pup and $\pi$ Aqr, we were allowed to set some constraints on the disc parameter values. Even though the synthetic spectra adequately fit our observations of both stars, the derived values show a discrepancy with the observed data in the EW and flux plots. The discrepancy is more remarkable in the case of MX Pup, where non-coherent electron scattering and a more complex radial dependence for the density seem to be needed.

It is possible that by including Brackett lines of higher terms or adding a similar analysis to the one performed throughout this article to other hydrogen series, we may be able to improve the determination of the parameters of the observed disc. We defer such analysis to a forthcoming article.

**Author Contributions:** Conceptualisation, Y.R.C., A.G., M.L.A. and A.F.T.; methodology, Y.R.C., A.G., M.L.A., A.F.T. and C.A.; formal analysis, Y.R.C., A.G., M.L.A. and A.F.T.; writing—original draft preparation, Y.R.C.; writing—review and editing, Y.R.C., A.G., M.L.A., A.F.T. and C.A.; visualisation, Y.R.C.; funding acquisition, A.G., M.L.A. and A.F.T. All authors have read and agreed to the published version of the manuscript.

**Funding:** The authors would like to thank the reviewers for all their useful comments, which improved our manuscript. Y.R.C. acknowledges support from a CONICET fellowship. M.L.A. and A.F.T. acknowledge financial support from the Universidad Nacional de La Plata (Programa de Incentivos 11/G160). A.G. acknowledges the research project from CONICET, PIBAA 28720210100879CO. C.A. thanks Fondecyt Regular N. 1230131 for the support. This project has received funding from the European Union's Framework Programme for Research and Innovation Horizon 2020 (2014–2020) under the Marie Skłodowska-Curie Grant Agreement No. 823734.

**Data Availability Statement:** The data presented in this study are available on request from the corresponding author.

**Acknowledgments:** This work made use of Astropy:[5] a community-developed core Python package and an ecosystem of tools and resources for astronomy [38–40]. The model calculations were performed with the cluster of CITECCA of Universidad Nacional de Río Negro, Argentina, which has four processors with eight threads, 2.2 GHz speed, and 180 GB of RAM. This paper includes data gathered with the 6.5 m Magellan Telescopes located at Las Campanas Observatory, Chile.

**Conflicts of Interest:** The authors declare no conflict of interest.

## Abbreviations

The following abbreviations are used in this manuscript:

| | |
|---|---|
| EW | Equivalent width |
| IR | Infrared |
| LTE | Local thermodynamic equilibrium |
| SED | Spectral energy distribution |

## Appendix A

*Appendix A.1. Synthetic Brackett Line Profiles Obtained for $i = 30°$ and $i = 60°$*

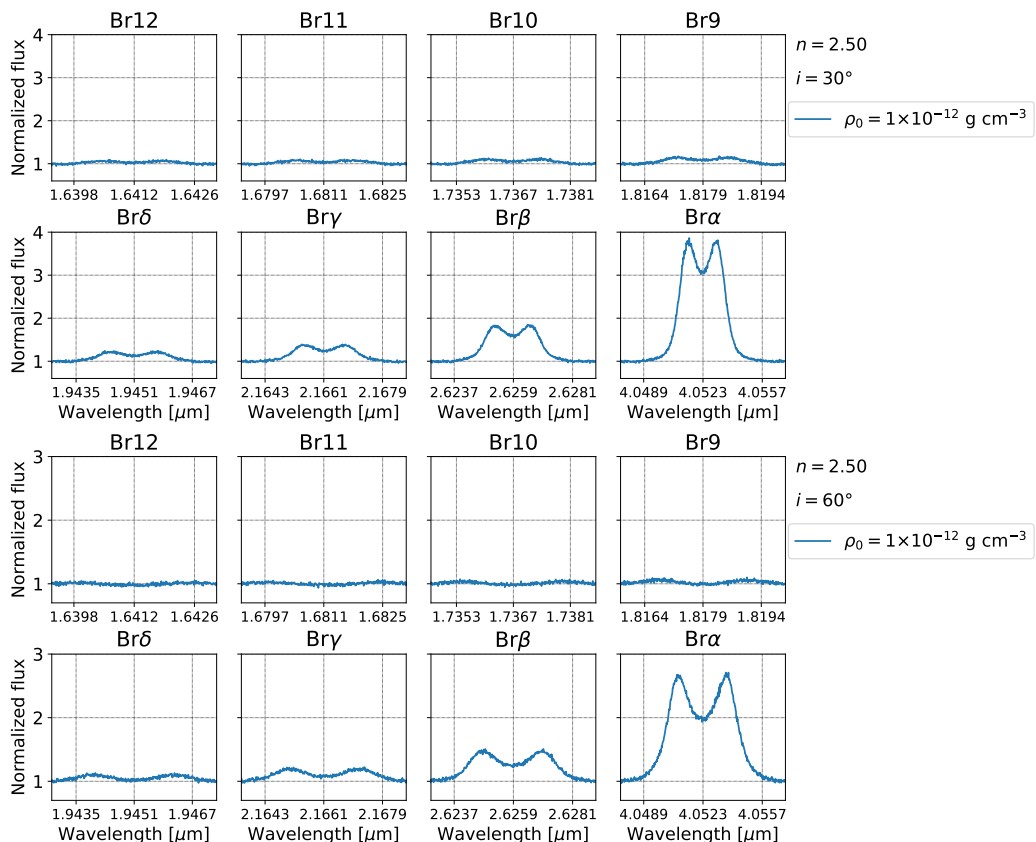

**Figure A1.** Synthetic Brackett line profiles obtained for an exponent of the density law $n = 2.5$, $\rho_0 = 1 \times 10^{-12}$ g cm$^{-3}$, and inclinations $i = 30°$ and $i = 60°$.

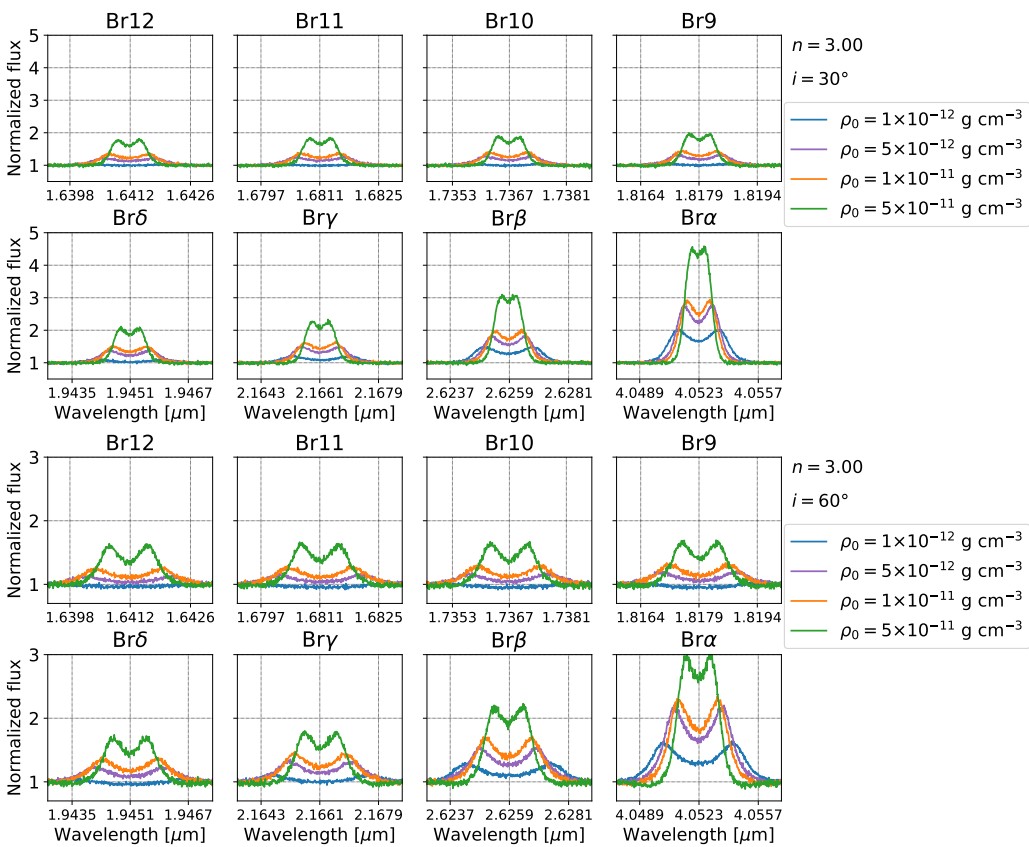

**Figure A2.** Synthetic Brackett line profiles obtained for an exponent of the density law $n = 3.0$, different values of the central density, and inclinations $i = 30°$ and $i = 60°$.

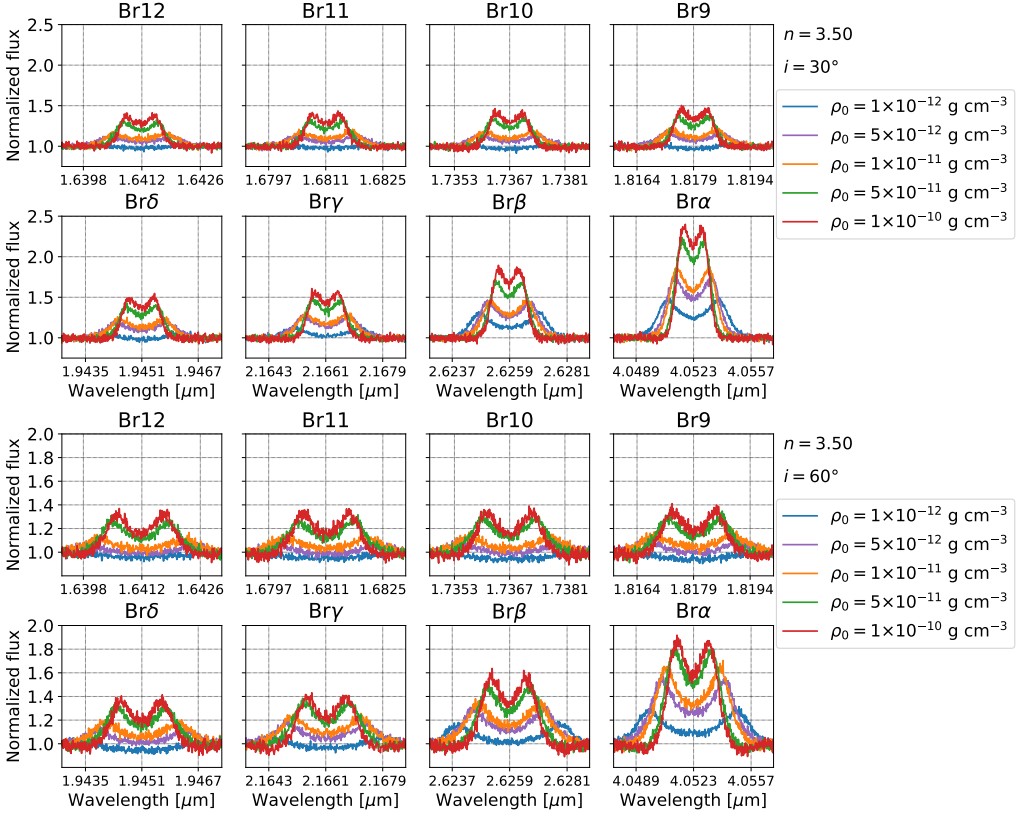

**Figure A3.** Synthetic Brackett line profiles obtained for an exponent of the density law $n = 3.5$, different values of the central density, and inclinations $i = 30°$ and $i = 60°$.

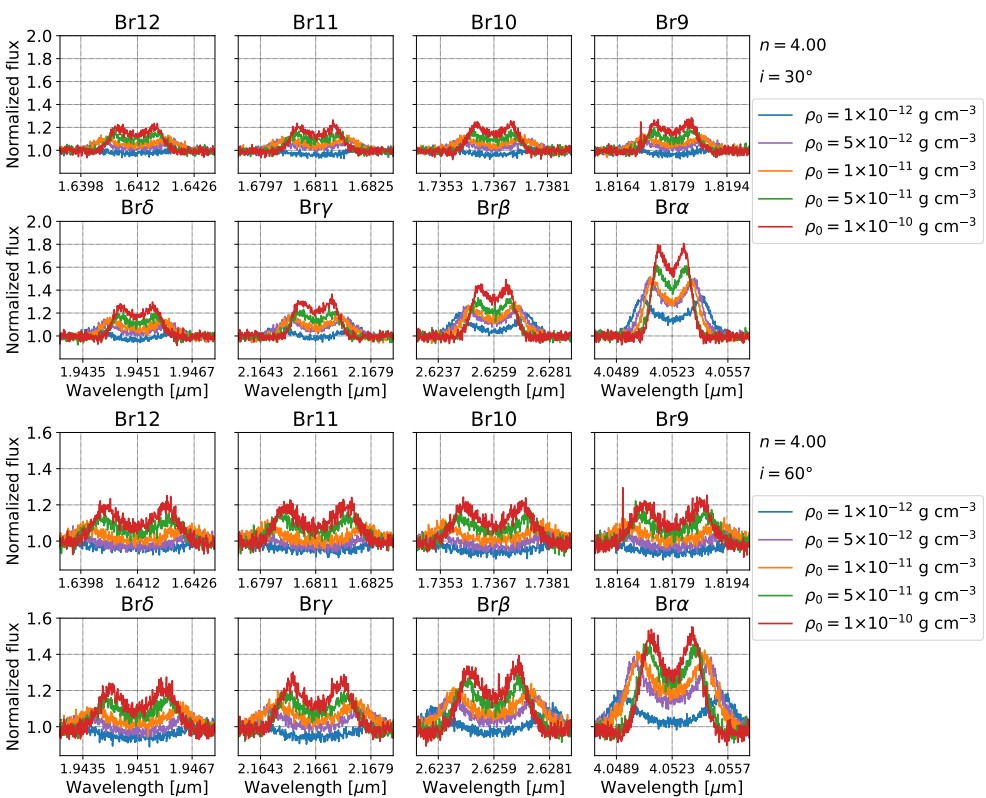

**Figure A4.** Synthetic Brackett line profiles obtained for an exponent of the density law $n = 4.0$, different values of the central density, and inclinations $i = 30°$ and $i = 60°$.

*Appendix A.2. EW and Flux Ratios for $i = 30°$ and $i = 60°$*

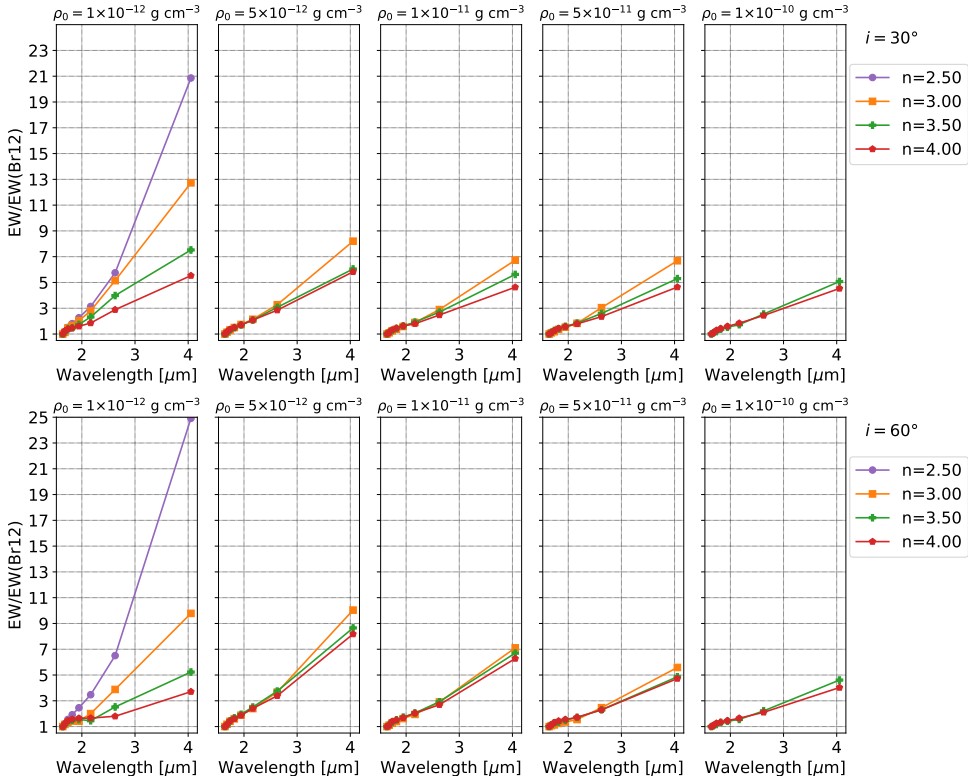

**Figure A5.** EW ratio for the Brackett lines relative to EW(Br12) for $i = 30°$ and $i = 60°$. Each panel corresponds to a given central density $\rho_0$, and each curve represents a given $n$ value.

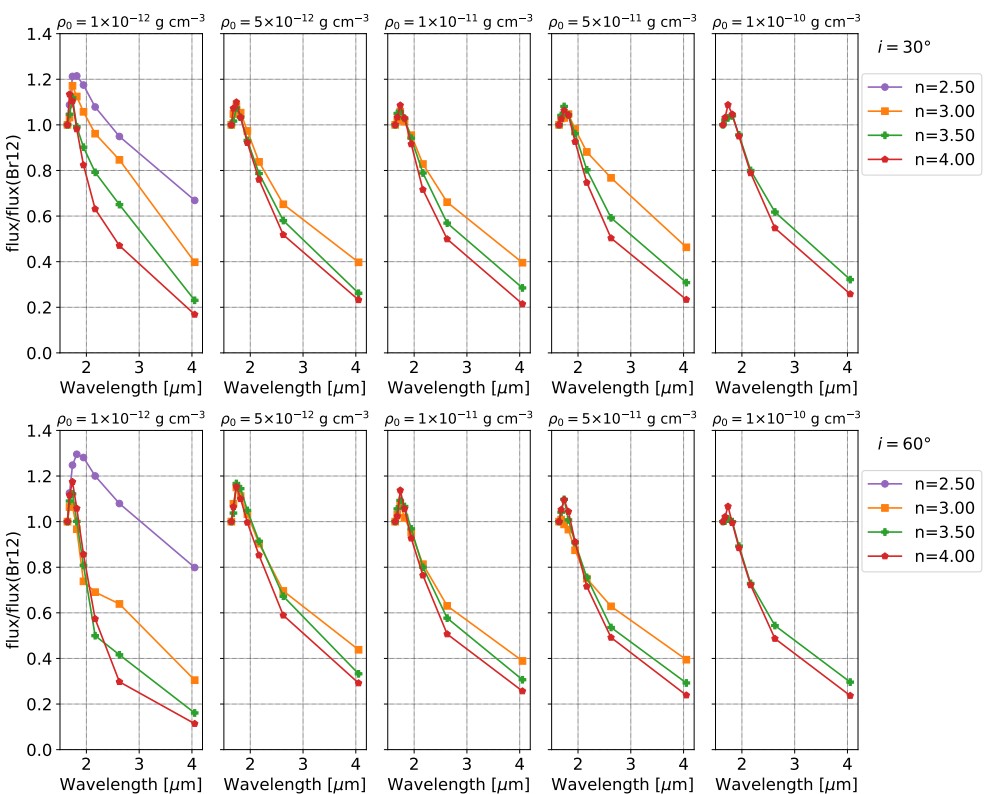

**Figure A6.** Flux ratio for the Brackett lines relative to flux(Br12) for $i = 30°$ and $i = 60°$. Each panel corresponds to a given central density $\rho_0$, and each curve represents a given $n$ value.

## Notes

[1] The code actually uses $n_0$, the density in numerical units per $cm^{-3}$. Both parameters are related via the expression $\rho_0 = n_0 \cdot \mu \cdot m_H$, where $\mu = 0.6$ is the mean molecular weight, and $m_H = 1.67 \times 10^{-24}$ g is the mass of the hydrogen atom. Then, $\rho_0 \simeq n_o \times 10^{-24}$ g.

[2] Apart from the models that we present in Table 1, we computed additional models for each to account for the disc's outer radius. We computed around 50 models, which took approximately 150 hours of calculation in CITECCA's cluster (see Acknowledgments for more information).

[3] PyHdust provides analysis tools for multi-technique astronomical data and HDUST models.

[4] Specutils is a Python package for representing, loading, manipulating, and analysing astronomical spectroscopic data [30].

[5] http://www.astropy.org, accessed on 15 August 2023.

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
