# Peer review of "Infrared Spectroscopy of Be Stars: Influence of the Envelope Parameters on Brackett-Series Behaviour"

_galaxies, doi:10.3390/galaxies11040090_

Round 1
Reviewer 1 Report
I have read the paper, "IR spectroscopy of Be stars: influence of the envelope parameters in the Brackett-series behaviour" by Cochetti et al. Due to a lack of content and other issues I find that the paper needs substantial revision before it can be accepted for publication.
The authors have run a small grid of Be disk models using the code HDUST. The grid varies two parameters — the initial disk density and the power law exponent that describes the variation of density with radius. One set of stellar parameters is used. The authors then study the variation of members of the hydrogen Bracket series across the models, before undertaking a brief comparison with observation.
My comments on the paper follow:
Insufficient details are supplied about the construction of the models. For example, how was the scale height computed? What is the inner disk radius? What assumptions were made in the model setup? I am presume HDUST has different parameter choices. How long did it take to compute each model? What criteria did they assume regarding model convergence?
Only one set of parameters were included. Are other models with different stellar parameters planned? What influence does the choice of stellar parameters have on disk spectra (presumably this will have been discussed in the extensive literature on Be stars)?
In line 72 (page 2) the authors state "Such value largely exceeds the formation region …". Largely is an imprecise word that does not convey any real meaning. How much does the choice of the outer radius affect the answers, especially at the highest densities? Did the authors check where the emission was originating from in the models? Did the authors run a model with an extended radius to confirm that the choice of outer radius was not influencing their results.
The paper lacks references. There is a large amount of literature on Be stars, and this paper leaves the impression that we are still struggling to understand Be stars. However, over the last 20 years there has been huge progress in modeling (including time-dependent modeling) and in observations (particularly interferometry) of Be stars. Much better referencing of the current literature is needed, especially that relevant to modeling Be disks. Even some of their own work is not adequately referenced.
Nearly half the abstract discusses the comparison with observation, but less than half a page of text is devoted to the comparison in the paper. Two Be stars were considered — 12 Vul and MX Pup. Virtually no details about these objects are provided to the reader. Why were these two stars chosen? Do the model stellar parameters match those of these two stars? What do we know already about these two stars and their disks?
In the comparison with observations the authors conclusions are not particularly exciting. For example, the limits they can set on the disk properties of 12 Vul are rather weak. Once they realized this the authors should have looked for other factors to constrain Be disk parameters. I presume much of the work in the model construction was in the non-LTE computation of the disk model — not the profile computation. What other lines could be used with the Brackett series to constrain Be disk parameters? What constraints do the infrared flux (or fluxes) provide when used in conjunction with the Brackett series? The authors only discuss line fluxes and EWs for the comparison — what about information from line profiles? The authors show ratios of B12 to B11 and, not surprisingly given the nature of the lines, their fluxes are highly correlated. Given their earlier plots, and given the similarity
of the curves shown in Fig 10, the ratio of B12/B11 could be specified by a single number with a small dispersion. Such a number is probably more useful than Fig. 10.
More minor comments
Figures: Most of the diagrams have fluxes/EWs plotted versus wavelength. Not sure this is the best choice as high members of the Bracket series cluster together. While some plots like this might be useful, other plots using the upper principal quantum for the x-axis might be clearer.
Supply reference for the PyHdust and specutils packages (and what do each of these packages do?)
English is generally fine. Given my statement that the paper needs substantial revision I do not provide minor English corrections in this report.
The authors motivation for doing this work seems uninspired. Why use the infrared? Are there reasons that the IR might be a useful tool for understanding Be stars?
Author Response
Dear reviewer,
We are resubmitting our paper. The article has significant changes with respect to the previous version, following your and other reviewers' recommendations. That really helped to improve our manuscript significantly, so I would like to thank you for all your valuable comments. All the changes are in bold font across the text.
I want to highlight some of the bigger changes we made:
- Introduction: We include more references regarding the actual knowledge on Be stars to properly introduce our topic. We hope this would help to clarify our main (future) objective and the scope of this work.
- Method: More information about the construction of the models was added. Regarding the disc size, the generic one R=20R* considered for the models was not correct for some of them. We have run more models to determine the minimum disc size needed for each one. For the models with low density or high exponent of the density law, the R=20R* we considered was enough to include all the emission regions. For others, we were indeed not taking into account part of the emission, especially for the first member of the series. We fixed that.
- Discussion: The stellar parameters of 12 Vul were actually not close enough to the model parameters to rely on the comparison. We keep the comparison with MX Pup and added Pi Aqr. We include more information about the stellar parameters and why we choose these objects. We also added a comparison with the spectrum of these objects.
Regarding some of your comments not included in the mentioned changes I would like to add the following:
- How long did it take to compute each model? What criteria did they assume regarding model convergence?
We added information about the consumed time for the simulations and about the computational resources used. The convergence criterium was also included in the text.
- Are other models with different stellar parameters planned?
Yes, we plan to increase our grid gradually. Our final aim is to use the grid to deduce disk properties, and this is just the first step. Then we only include one set of stellar parameters and only the first Brackett lines.
- The authors show ratios of B12 to B11 and, not surprisingly given the nature of the lines, their fluxes are highly correlated. Given their earlier plots, and given the similarity of the curves shown in Fig 10, the ratio of B12/B11 could be specified by a single number with a small dispersion. Such a number is probably more useful than Fig. 10.
The fluxes of the B12 and Br11 lines are indeed highly correlated, but the locus of each model on the plot depends on the disk parameters and this is what we wanted to show with this figure.
Best regards,
Yanina Cochetti
Reviewer 2 Report
This manuscript presents a grid of HDUST model predictions for IR Brackett lines appropriate to the properties of Be stars plus (brief) comparisons with spectroscopic observations of two Be stars from the authors.
Overall the paper focuses on line (flux/equivalent width) predictions for a range of densities and radial dependence, albeit with too little attention given to how disk characteristics can be obtained from comparisons with observations in practice. Balance between simulations and application is too heavily biased towards the former.
Observations of two Be stars are drawn from a large compilation by Cochetti+2022 using a mix of Gemini GNIRS (HKL bands for 12 Vul) and Magellan FIRE (JHK bands for MX Pup) instruments without explanation, since the present focus on Brackett series provides a mix of diagnostics (Br-alpha for 12 Vul only) and the adopted stellar properties (R_pole=5.5 Rsun, L=7500 Lsun, M=10 Msun) are *reasonable* for MX Pup (given its distance, temperature, gravity) but far from reasonable for 12 Vul (L~1000 Lsun given its distance, temperature).
The significance of the present paper would be greatly enhanced if the existing HDUST grid was exploited to *deduce* disk properties (densities, radial dependence, inclination) from a range of IR diagnostics of MX Pup (for adopted stellar properties), including line profile information (compare Fig 2 to A1) and Paschen to Brackett ratios.
Eqn 1 shows density structure, incl rho_0, but grid focuses on n_0 with an "approximate" factor of 1e24 between the two quantities (shouldn't factor be 1.67e24 for a pure hydrogen disk)
Group I Be stars is utilised in Section 3 but not defined (cite Lenorzer+2002).
Author Response
Dear reviewer,
We are resubmitting our paper. The article has significant changes with respect to the previous version, following your and other reviewers' recommendations. That really helped to improve our manuscript significantly, so I would like to thank you for all your valuable comments. All the changes are in bold font across the text.
I want to highlight some of the bigger changes we made:
- Introduction: We include more references regarding the actual knowledge on Be stars to properly introduce our topic. We hope this would help to clarify our main (future) objective and the scope of this work.
- Method: More information about the construction of the models was added. Regarding the disc size, the generic one R=20R* considered for the models was not correct for some of them. We have run more models to determine the minimum disc size needed for each one. For the models with low density or high exponent of the density law, the R=20R* we considered was enough to include all the emission regions. For others, we were indeed not taking into account part of the emission, especially for the first member of the series. We fixed that.
- Discussion: The stellar parameters of 12 Vul were actually not close enough to the model parameters to rely on the comparison. We keep the comparison with MX Pup and added Pi Aqr. We include more information about the stellar parameters and why we choose these objects. We also added a comparison with the spectrum of these objects.
Regarding some of your comments not included in the mentioned changes I would like to add the following:
- The significance of the present paper would be greatly enhanced if the existing HDUST grid was exploited to *deduce* disk properties (densities, radial dependence, inclination) from a range of IR diagnostics of MX Pup (for adopted stellar properties), including line profile information (compare Fig 2 to A1) and Paschen to Brackett ratios.
Our final aim is to use the grid to deduce disk properties, and this is just the first step. Then we only include one set of stellar parameters and only the first Brackett lines. We include now the line profile comparison for both MXPup and PiAqr, but the Paschen to Brackett ratios exceed the scope of this work.
- Eqn 1 shows density structure, incl rho_0, but grid focuses on n_0 with an "approximate" factor of 1e24 between the two quantities (shouldn't factor be 1.67e24 for a pure hydrogen disk)
The relation between rho_0 and n_0 has been included. Nevertheless, for simplicity, we choose to transform the n_0 to rho_0 values across the text.
- Group I Be stars is utilised in Section 3 but not defined (cite Lenorzer+2002).
We added the correspondent reference and explanation.
Best regards,
Yanina Cochetti
Reviewer 3 Report
I have read the manuscript entitled “IR spectroscopy of Be stars: influence of the envelope parameters in the Brackett-series behaviour” by Yanina R. Cochetti, Anahi Granada, Maria Laura Arias, Andrea F. Torres, and Catalina Arcos with great interest. The manuscript introduces infrared synthetic model spectra computed for a range of parameters (central number density, exponent of the density profile, and inclination). After discussing the trends in this parameter space in terms of the Brackett series line profiles, equivalent widths, and fluxes, the models are compared with observations of two Be stars. The Authors conclude that specific values for the disc parameters cannot yet be inferred from a degenerate parameter space.
I find the manuscript informative and overall well-written. I recommend it for publication after minor revision.
*******************
Main comments:
*******************
Line 76: The central star is assumed to be a rigid rotator. Could the Authors comment on this assumption and its possible consequences? For example, radial and latitudinal differential rotation is inferred in some stars.
Lines 155-157: It is not trivial to compare by eye how well the obtained trend matches the linear regression of Steele and Clark. Could the Authors please quantify the linear regression they find and compare it directly with the linear fit expected for Case B recombination?
Could the Authors comment on the mass-loss rates in their models and include them in the comparison with observations?
Figures 11-14: Could the Authors include the uncertainty associated with the observations on these diagrams and also mention it explicitly in the text? (For the observations, higher than Br12 lines are also shown with markers, which is potentially confusing here since the models are calculated up to Br12.)
The Authors find that some constraints may be placed on the m parameter in the case of MX Pup but not in the case of 12 Vul. I believe this might partially result from the lack of Bralpha measurements in the former case. In the case of 12 Vul, the measurements point to a discrepancy to explain both n_0 and m values simultaneously. In Figure 12, the Authors find m=3.5 and n_0 =< 10^13 cm^-3 to be good matches for 12 Vul. The same values; however, are not unambiguous in Figure 11. For example, at n_0=5x10^12 cm^-3, EW(Bralpha) implies m < 2.5, and Fl(Bralpha) implies m < 2.5 (or m > 4). This is at odds with the results from the Br11–Br12 equivalent widths, which indicate approximately 3 < m < 4.
For the same reason, I would caution against an interpretation solely based on the Br11–Br12 equivalent widths in the case of MX Pup. At least, from Figure 13 it seems that constraints cannot be established at this stage.
*******************
Minor comments:
*******************
Some authors and readers might prefer avoiding acronyms or abbreviations in the title, abstract, and section title.
Table 1: Since inclination is also a parameter varied in the models, I would recommend making Table 1 an overview of all the computed models and listing the inclination angles. In this table, I would also strongly suggest highlighting the different disc types as a function of the exponent of the density profile.
Below Equation 1: for completeness, could the Authors please add the definition of n_0? Perhaps also the expression of how the central number density is converted to mass density? Throughout the text, “density” is often used but it refers to n_0, whereas in Equation 1 only the mass density is introduced. Some potential confusion might arise from this. Therefore I would suggest revising the text and being more specific where “density” is mentioned.
Figure 1: I would suggest explicitly stating what inclination angle the pole-on orientation refers to in the chosen reference frame. Since this figure serves as a demonstration, perhaps could it be useful to label each individual Brackett line too?
Figures 2-8: Since the explanations are discussed in the main text, I would suggest shortening the captions to keep only the description of the figures. The discussion of the trends is now duplicated in the text and in the captions.
Line 153: For clarity, could the Authors please state explicitly their definition of Group I stars (EW(Bralpha) < 3 * EW(Brgamma))?
*******************
While I find the manuscript well-written and comprehensible, there are instances where the language/style could perhaps be improved. I list a few of these here along with my recommendations.
Line 40: outcoming spectra -> emerging spectra
Line 62: density on the base of the disc -> density at the base of the disc
Line 65-67: We computed those models marked with a [tick mark] that indicate those combinations of parameters that are typically adequate to fit Be-star Halpha profiles. ->
1) We computed those models marked with a [tick mark] that indicate combinations of parameters typically adequate to fit Be-star Halpha profiles.
2) We only computed models for combinations of n_0 - m parameters that are typically adequate to fit Be-star Halpha profiles. These are marked with a [tick mark].
Line 98: increase on the intensity -> 1) increase in intensity 2) increase of the intensity
Lines 149-150: Similarly than the mentioned for the EW -> Similarly to the EW, the flux [shows no significant change]
Lines 185-186: The models we used have been found suitable to fit Be-star Hα profiles [e.g. 15]. -> The parameter space of the models we used has been found suitable to fit Be-star Hα profiles [e.g. 15].
Finally, the spelling of some words seems to vary (normalized - normalised, analyzed - analysed). I would suggest keeping overall consistency in this regard.
Author Response
Dear reviewer,
We are resubmitting our paper. The article has significant changes with respect to the previous version, following your and other reviewers' recommendations. That really helped to improve our manuscript significantly, so I would like to thank you for all your valuable comments. All the changes are in bold font across the text.
I want to highlight some of the bigger changes we made:
- Introduction: We include more references regarding the actual knowledge on Be stars to properly introduce our topic. We hope this would help to clarify our main (future) objective and the scope of this work.
- Method: More information about the construction of the models was added. Regarding the disc size, the generic one R=20R* considered for the models was not correct for some of them. We have run more models to determine the minimum disc size needed for each one. For the models with low density or high exponent of the density law, the R=20R* we considered was enough to include all the emission regions. For others, we were indeed not taking into account part of the emission, especially for the first member of the series. We fixed that.
- Discussion: The stellar parameters of 12 Vul were actually not close enough to the model parameters to rely on the comparison. We keep the comparison with MX Pup and added Pi Aqr. We include more information about the stellar parameters and why we choose these objects. We also added a comparison with the spectrum of these objects.
Regarding some of your comments not included in the mentioned changes I would like to add the following:
- Could the Authors comment on the mass-loss rates in their models and include them in the comparison with observations?
We added in the model description what would be the mass-loss rate for each density. It is worth mentioning that this mass-loss rate is valid for a disc in a steady state and should be taken into account as estimations.
- Could the Authors include the uncertainty associated with the observations on these diagrams and also mention it explicitly in the text? (For the observations, higher than Br12 lines are also shown with markers, which is potentially confusing here since the models are calculated up to Br12.)
We added the uncertainties for EW and flux measurements both in the text and the graphics, and only keep the measurement for the modelled lines.
- Below Equation 1: for completeness, could the Authors please add the definition of n_0? Perhaps also the expression of how the central number density is converted to mass density? Throughout the text, “density” is often used but it refers to n_0, whereas in Equation 1 only the mass density is introduced. Some potential confusion might arise from this. Therefore I would suggest revising the text and being more specific where “density” is mentioned.
The relationship between rho_0 and n_0 has been included. Anyway, for simplicity, we chose to transform the values n_0 to rho_0 throughout the text.
Best regards,
Yanina Cochetti
Round 2
Reviewer 1 Report
I have read the paper, "IR spectroscopy of Be stars: influence of the envelope parameters in the Brackett-series behavior" by Cochetti et al. The authors have made a substantial effort to improve the paper and I believe it should be published. However I do have some additional suggestions the authors should consider before the paper is formally accepted.
Science comments
L159 — unsure as to the meaning of intensities — need to be more specific. I presume you mean the height of the line relative to its adjacent continuum (I/I_c).
On the figures showing line profile, a slightly smaller font for the axis labels might help the readability of the abscissa scale. (which otherwise are very clear)
Does HDUST include non-coherent electron scattering and could this explain the wings on the Bracket lines in Fig. 10?
The Br-gamma line for pi Aqr does not show the same profile issues, so perhaps worth a comment..
As the authors note, the fluxes in MX Pup are larger than predicted. Is this primarily due to the emission in the wings?
In Fig 10, using velocity for the abscissa scale would be more meaningful.
Did you compare predicted and observed continuum fluxes?
Line 216 — 221. Is MX Pup a variable? A different rotational velocity seems unlikely since this would simply broaden the profile. The profile in Fig. 10 for Br-gamma seems to indicate a distinct emission region for the line wings. The central part of the profile is well matched by the model. Do any of your models show similar profiles to that seen in MX Pup? How do the profiles of Br-gamma and Br12 compare in velocity space.
In reading your abstract I have the impression that there is a general disagreement between the models and observations, but from the figures later on in the text, the main problem is with MX Pup, and the agreement for pi Aqr is much better.
English comments
The english is readable, but it would help if someone more familiar with English went though the manuscript with the authors to improve the grammar.
Below is my suggestion for an improved abstract (that does not change its meaning):
The IR spectra of Be stars display numerous hydrogen recombination lines, constituting a great resource for obtaining information on the physical and dynamic structure of different regions within the circumstellar envelope. Nevertheless, this spectral region has not been analysed in depth, and there is a lack of synthetic spectra to compare with available spectroscopic observations. Therefore we computed synthetic spectra with the HDUST code for different disc parameters. Here, we present our results on the spectral region that includes lines of the Brackett series. We discuss the dependence of the line series strengths on several parameters that describe the structure of the disk. We also compared model line profiles, fluxes, and EWs with observational data for two Be stars (MX Pup 8 and π Aqr). Even though the synthetic spectra adequately fit our observations of both stars, and allow us to constrain the parameters of the disc, there is a discrepancy with the observed data in the EWs and fluxes measurements. It is possible that by including Brackett lines of higher terms or adding the analysis of other series, we may be able to better constrain the parameters of the observed.
Some additional English suggestions:
L18 — already accepted —> accepted
L20 — better explains —> best explains
L23 — their spectra —> Be spectra
L43 — works focussed —> works that focussed (remove , after Be stars)
L66 — The modelling of different observables, and in particular the most iconic tracer of Be star discs, the Hα line emission, has allowed constraining the Be disc parameters in the past years, indicating that this line typically forms within the innermost tens of stellar radii —> The modelling of different observables has allowed constraints on Be disc parameters to be derived. In particular, modeling of the iconic Hα line emission Ha showed that this line typically forms within the innermost tens of stellar radii
L71 — is of at most —> is at most
L79 the first lines—> lines
L108 gr —> g
L 173: — We corrected the measured equivalent width (EW) from the emission lines, considering the underlying photospheric absorption, to have the total contribution from the disc —> To deduce the contribution of the disc we corrected the measured emission line equivalent widths (EWs) for photospheric absorption
L199: For such analysis, we have chosen two stars with stellar parameters close to our modelling from the published sample in Cochetti et al. [12]: MX Pup and π —> Using the published sample in Cochetti et al. [12] we have chosen two stars with stellar parameters close to that used in our modelling: MX Pup and π Aur.
L208: to consider —> to match
L212: — For MX Pup, Fremat et al,
L219 — It is worth mentioning that with this — With this
Author Response
Dear reviewer,
I include here some comments regarding your last revision. Thank you again for your valuable suggestions.
Best regards,
Yanina Cochetti
L159 — unsure as to the meaning of intensities — need to be more specific. I presume you mean the height of the line relative to its adjacent continuum (I/I_c).
Yes, we mean the I/Ic value. We changed it in the text.
On the figures showing line profile, a slightly smaller font for the axis labels might help the readability of the abscissa scale. (which otherwise are very clear)
We modified the x-axis labels.
Does HDUST include non-coherent electron scattering and could this explain the wings on the Bracket lines in Fig. 10? The Br-gamma line for pi Aqr does not show the same profile issues, so perhaps worth a comment..
No, HDUST does not include non-coherent electron scattering and that could affect the line profile. We added some comment about that, and how that could be improved in the future.
As the authors note, the fluxes in MX Pup are larger than predicted. Is this primarily due to the emission in the wings?
Yes, the difference in the wings can be the reason of our discrepancies.
In Fig 10, using velocity for the abscissa scale would be more meaningful.
We changed the plots.
Did you compare predicted and observed continuum fluxes?
No, we didn’t. We may include this analysis in future work.
Line 216 — 221. Is MX Pup a variable? A different rotational velocity seems unlikely since this would simply broaden the profile. The profile in Fig. 10 for Br-gamma seems to indicate a distinct emission region for the line wings. The central part of the profile is well matched by the model. Do any of your models show similar profiles to that seen in MX Pup?
Yes, its light curve presents long-term variability. It is possible that to assume a single power-law it is not enough to explain the complexity of the disc, especially if the disc has contribution of different mass loss events. We added a comment about that possibility.
Some models seems to fit better just the more extended wings.
How do the profiles of Br-gamma and Br12 compare in velocity space.
The peak separation for Br12 is larger that the one for Brgamma, as it is expected for higher member of the series forming in a region closer to the star. We added this to the text.
In reading your abstract I have the impression that there is a general disagreement between the models and observations, but from the figures later on in the text, the main problem is with MX Pup, and the agreement for pi Aqr is much better.
We added a phrase in the abstract trying to clarify this.
Reviewer 2 Report
I would like to thank the authors for considering my previous suggestions for the manuscript. I have a few remaining points to consider, all relatively minor
Scientific:
Since Be stars are believed to be rapid rotators, please specify R_eq and R_p for equatorial and polar radius (latter is already done) in preference to R_{\ast} for stellar radius. From critical velocity definition it would appear R_{\ast} = R_eq.
Although the emphasis of this paper is on a grid of models, the comparisons with Be star observations in Section 4 is a welcome addition (especially Figs 10 and 13). Some additional comments are necessary in determining the best fitting model given the relative insensitive of EqW and flux ratios to density and n presented in Figs 11 and 14. Although the agreement achieved for pi Aql (Fig 13) is rather good, there are significant discrepancies for MX Pup (Fig 10) regarding line width and separation of blue+red disk components (also relevant for Fig 12) requiring additional comments beyond the brief mention in Sect 4.1.
For the present grid the adopted stellar model has v_rot = v_eq = 370 km/s, approx 3/4 of the critical (max) rotational rate for a 10 solar mass star. The adopted v_eq * sin i for the two case studies coupled with the preferred inclinations from your analysis range from half critical (MX Pup) to near critical (pi Aql). Could you comment on whether large deviations from near critical rotation are physically realistic for Be systems?
Non-scientific:
Abstract "to compare the available spectroscopic observations with" -> "with which to compare to observations"
"with FIRE spectrograph from Las Campanas Observatory" -> "with the FIRE spectrograph installed at the Magellan Baade Telescope at Las Campanas Observatory"
Author Response
Dear reviewer,
I include here some comments regarding your last revision. Thank you again for your valuable suggestions.
Best regards,
Yanina Cochetti
Since Be stars are believed to be rapid rotators, please specify R_eq and R_p for equatorial and polar radius (latter is already done) in preference to R_{\ast} for stellar radius. From critical velocity definition it would appear R_{\ast} = R_eq.
We clarified this point.
Although the emphasis of this paper is on a grid of models, the comparisons with Be star observations in Section 4 is a welcome addition (especially Figs 10 and 13). Some additional comments are necessary in determining the best fitting model given the relative insensitive of EqW and flux ratios to density and n presented in Figs 11 and 14. Although the agreement achieved for pi Aql (Fig 13) is rather good, there are significant discrepancies for MX Pup (Fig 10) regarding line width and separation of blue+red disk components (also relevant for Fig 12) requiring additional comments beyond the brief mention in Sect 4.1.
We added more information about how be determined our best-fit model, adding also possible reasons for the discrepancies we found.
For the present grid the adopted stellar model has v_rot = v_eq = 370 km/s, approx 3/4 of the critical (max) rotational rate for a 10 solar mass star. The adopted v_eq * sin i for the two case studies coupled with the preferred inclinations from your analysis range from half critical (MX Pup) to near critical (pi Aql). Could you comment on whether large deviations from near critical rotation are physically realistic for Be systems?
There are some studies that claims that Be stars do not rotate critically, but presents a wide range of true velocity ratios. For example, in Zorec et al. 2016, the found V/Vc values goes from 0.3 to 0.95.